# Measuring and Controlling Split Layer Privacy Leakage Using Fisher Information

**Kiwan Maeng**
Pennsylvania State University
kvm6242@psu.edu

**Chuan Guo**
Meta AI
chuanguo@meta.com

**Sanjay Kariyappa**
Georgia Institute of Technology
sanjaykariyappa@gatech.edu

**G. Edward Suh**
Meta AI / Cornell University
edsuh@meta.com

## Abstract

Split learning and inference propose to run training/inference of a large model that is split across client devices and the cloud. However, such a model splitting imposes privacy concerns, because the activation flowing through the split layer may leak information about the clients' private input data. There is currently no good way to quantify how much private information is being leaked through the split layer, nor a good way to improve privacy up to the desired level.

In this work, we propose to use Fisher information as a privacy metric to measure and control the information leakage. We show that Fisher information can provide an intuitive understanding of how much private information is leaking through the split layer, in the form of an error bound for an unbiased reconstruction attacker. We then propose a privacy-enhancing technique, ReFIL, that can enforce a user-desired level of Fisher information leakage at the split layer to achieve high privacy, while maintaining reasonable utility.

## 1 Introduction

Private-input split learning [1] and split inference [2] propose to run training/inference of only the first few layers of a model on clients' devices and run the rest on a server. These approaches are gaining interest as a way to run large, modern ML models [3, 4, 5, 6] that cannot fit on a single client device, without the clients having to share their private input data. Instead of sharing raw data, clients share the activation of the last layer of the client-side model (i.e., *split layer*).

However, recent studies showed that only sharing this split layer activation still leaks information about the clients' private input, often significantly enough to reconstruct the original input precisely [7, 8, 9, 10] (Figure 1a). There has not been a good understanding of how much information leaks through the split layer activation, or how one can reduce such an information leakage. For the body of proposals of split learning and inference to become practical, it is crucial to have (1) a **theoretically-meaningful privacy metric** that can quantify the information leakage through the split layer activation, and (2) a **controllable privacy-enhancing method** that can reduce the amount of leakage arbitrarily. All prior works that aimed to quantify or improve the privacy of the split layer activation failed to meet one or both of the above criteria [1, 11, 12, 13, 14, 15, 16, 17, 18, 19, 20].

For the first time to the best of our knowledge, we propose a metric and a privacy-enhancing method that meets both above criteria, in the context of preventing input reconstruction attacks on a split inference setup (Figure 1b). For the theoretically-meaningful privacy metric, we propose to use

Workshop on Federated Learning: Recent Advances and New Challenges, in Conjunction with NeurIPS 2022 (FL-NeurIPS'22). This workshop does not have official proceedings and this paper is non-archival.

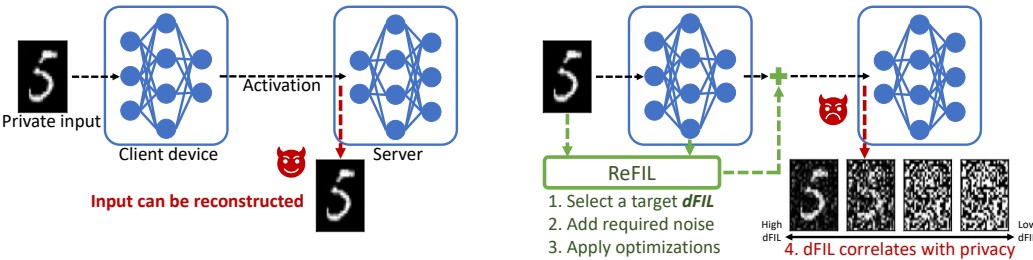

(a) Split layer activation leaks private input      (b) Proposed solution: dFIL and ReFIL

Figure 1: Overview of (a) the threat model the paper assumes and (b) our proposed solution.

*diagonal Fisher information leakage* (dFIL), a metric originally proposed to quantify training data leakage through model parameters [21]. With proper repurposing, we show that dFIL can quantify the information leakage through the split layer activation, by theoretically bounding the average reconstruction error against an unbiased reconstruction attack (Section 3). We subsequently propose a privacy-enhancing method, *ReFIL* (**Re**ducing d**FIL**), that can arbitrarily control the dFIL of a split layer while maintaining reasonable utility (Section 4). dFIL and ReFIL may be applicable to other split-model setups (e.g., split learning) and attacks. We leave further exploration as future work.

Evaluating against input reconstruction attacks [7, 8, 9] on a split inference setup, we demonstrate that dFIL correlates well with the reconstructed image quality, serving as a good privacy metric. We also show that with several optimizations, ReFIL can provide a practical privacy-utility trade-off. Below summarizes our contribution:

- We propose to use dFIL as a privacy metric for split layer activation. We theoretically and empirically show that dFIL serves as a good privacy metric against input reconstruction attacks.
- We propose ReFIL, a privacy-enhancing method that controls the information leakage of a split layer activation while maintaining reasonable utility through several optimizations.
- We evaluate the efficacy of dFIL and ReFIL on a split inference setting, using popular image (MNIST [22], CIFAR-10 [23]) and recommendation (MovieLens-20M [24]) datasets, popular models (ResNet-18 [25], NCF-MLP [26]), and different input reconstruction attacks [7, 9].

## 2   Background and Motivation

**Split inference and learning**    Split inference splits a model and runs the first few layers on client devices, while running the rest on a shared server [27, 2, 28, 29, 30, 31, 32]. Split inference allows running a large model without the client needing to share its private input. Instead, the client shares the activation of the split layer (Figure 1a). Similarly, split learning [1, 11] cuts and places a model across client devices and the server. Depending on what client data is considered sensitive, the client device may run the first few layers and share the split layer activation (private input), run the last few layers and share the split layer gradient (private label), or both [1]. When multiple clients participate, they can participate sequentially [1, 11, 12] or in parallel in a FL-like fashion [12].

**Information leakage through a split layer**    During split inference/learning, clients' private input [7, 8, 9, 10, 33] and label [34] can be reconstructed from the split layer activation/gradient, even when the client-side model is unknown [9, 10]. Among the attacks, we focus on input reconstruction attacks [7, 8, 9, 33, 10], which aims to reconstruct the private input from the split layer activation. Formally, let $\mathbf{z} = \mathcal{M}_{client}(\mathbf{x})$ be the split layer activation obtained by running an input $\mathbf{x}$ through a client-side model $\mathcal{M}_{client}$. An input reconstruction attack $\mathrm{Att}$ outputs an estimate of $\mathbf{x}$ by observing $\mathbf{z}$, $\hat{\mathbf{x}} = \mathrm{Att}(\mathbf{z})$. Such an attack is applicable to split inference and private-input split learning.

An input reconstruction attack is said to be *unbiased* if the reconstructed inputs' expected value is the same with the true input, i.e., $\mu(\mathbf{x}) = \mathbb{E}_{\mathcal{M}_{client}}[\hat{\mathbf{x}}]$. Many real-world attacks use prior of the input to improve the attack (e.g., encouraging low-frequency signal for images [7] or learning the distribution of the training data to generate in-distribution data [33]). These attacks are usually *biased*.

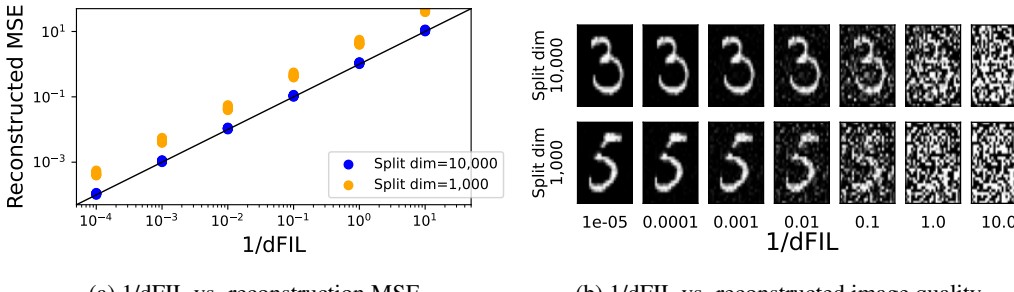

(a) 1/dFIL vs. reconstruction MSE

(b) 1/dFIL vs. reconstructed image quality

Figure 2: For an unbiased attack for an image model, (a) 1/dFIL lower-bounds the input reconstruction error, (b) directly controlling the reconstructed image quality.

**Threat model and goals**    In this work, we focus on a split inference setup, where the model up to the split layer runs on the client device, and rest runs on the server. We assume an honest-but-curious server as an attacker, who tries to reconstruct the input but does not alter the protocol in an unexpected way. We concentrate on input reconstruction attacks (Figure 1a) as they hamper clients' privacy most critically. We leave other attacks (e.g., property inference [35]) as future work. We assume the attacker knows the client-side model, which is realistic when the server is the model provider [27], is the model aggregator [12], or colludes with a client in a scenario like [12].

## 3    Quantifying the Information Leakage Through a Split Layer with dFIL

We propose to use dFIL to quantify the information leakage through the split layer in a split inference setup. As dFIL was originally for a different context [21], we present a brief repurposed formulation.

**Fisher information matrix (FIM) [36]**    FIM was originally proposed by [36] to quantify how much information a trained model holds about the training data. Here, we restate the definition modified for split inference. For a randomized method $\mathcal{A}(\mathbf{x}) = \mathcal{M}_{client}(\mathbf{x}) + \mathcal{N}(0, \sigma^2)$, the FIM of $\mathbf{z}' = \mathcal{A}(\mathbf{x})$ is defined as $\mathcal{I}_{z'}(\mathbf{x}) = \frac{1}{\sigma^2}\mathbf{J}_{\mathcal{M}_{client}}^T\mathbf{J}_{\mathcal{M}_{client}}$, where $\mathbf{J}_{\mathcal{M}_{client}}$ is the Jacobian of $\mathcal{M}_{client}$. FIM can be calculated only when a random noise $\mathcal{N}(0, \sigma^2)$ is added to the split layer.

**Diagonal Fisher information leakage (dFIL) [21]**    Subsequent work [21] showed that when an unbiased attacker tries to reconstruct the training data from the trained model, the average lower bound of the reconstruction error can be calculated using FIM. Using a repurposed formulation, we can show that the input reconstruction error can be bounded in split inference as well. For an unbiased reconstruction attack $\hat{\mathbf{x}} = \text{Att}(\mathbf{z}')$, [21] shows that the average mean-square error (MSE) of $\hat{\mathbf{x}}$ is bounded by:

$$\mathbb{E}[||\hat{\mathbf{x}} - \mathbf{x}||_2^2] \geq d/\text{Tr}(\mathcal{I}_{z'}(\mathbf{x})) \tag{1}$$

where $d$ is the dimension of $\mathbf{x}$ and Tr is the trace of a matrix. Here, $\text{Tr}(\mathcal{I}_{z'}(\mathbf{x}))/d$ is called *diagonal Fisher information leakage*, or dFIL. Proof can be adopted from [21].

**Using dFIL as a privacy metric for split inference**    dFIL is a strong indicator of privacy when it comes to input reconstruction attacks for split inference. For an unbiased attacker, dFIL directly indicates the reconstruction feasibility, as the average reconstruction error is bounded by 1/dFIL (Equation 1). For example, if the pixel value of an image is between [0, 1], and if dFIL=1, it is clear that the system will be nearly perfectly-private against an unbiased attacker, since the reconstruction error will be the same or larger than 1, which is no better than randomly choosing between [0, 1].

We demonstrate that the average reconstruction error of the unbiased attacker is indeed bounded by 1/dFIL, by running a multi-layer perceptron (MLP) with split inference and running an unbiased reconstruction attack against the setup. We split the model right after the first linear layer, varying the split layer dimension to be either 1,000 or 10,000. We used MNIST [22] dataset. More details about the setup and the attacker can be found in Section 5.1.

Figure 2a plots the resulting reconstruction mean-squared error (MSE) against 1/dFIL. Clearly, for an unbiased attacker, *the reconstruction MSE is lower-bounded by 1/dFIL*. Even when the activation

dimension is much larger than the original input (MNIST's input dimension is 28×28), the attacker was not able to bring down the reconstruction error below 1/dFIL. Figure 2b shows a randomly-selected reconstructed image for various dFIL. As expected, as the pixel values are between [0, 1], the system becomes perfectly private when 1/dFIL ≥ 1.

Many realistic attackers are biased, as they leverage prior knowledge of the input data [7, 8, 9, 33]. Still, we claim that dFIL can serve as a practical privacy metric for the biased attackers as well, as naturally, *inputs that are harder to reconstruct for an unbiased attacker will be harder to reconstruct in general, regardless of the attack method*. In Section 5.1, we empirically show that dFIL shows a strong correlation with reconstruction hardness even for biased attackers with a strong prior. It is unclear (although probable) if dFIL can serve as a good privacy metric for attacks other than input reconstruction (e.g., property inference [35]). We leave the discussion to future work.

## 4 ReFIL: Enforcing Desired dFIL with Practical Utility

We propose ReFIL (**Re**ducing d**FIL**) as a privacy-enhancing method that enforces the targeted dFIL to a split inference system while maintaining reasonable utility. From Section 3, it can be seen that enforcing a specific dFIL can be done by adding a Gaussian noise with $\sigma = \sqrt{\mathrm{Tr}(\mathbf{J}_{\mathcal{M}_{client}}^T \mathbf{J}_{\mathcal{M}_{client}})/(d * \mathrm{dFIL})}$ to the split layer activation. However, achieving smaller dFIL (better privacy) requires adding a larger noise, which can, in turn, hurt the utility of the system. ReFIL employs two optimizations that can reduce the noise that needs to be added while achieving the same dFIL, to explore a better privacy-utility trade-off.

**Optimization 1: compression/decompression layer** The first optimization drops unnecessary information flowing through the split layer by adding a compression layer at the end of the client-side model and a decompression layer at the beginning of the server-side model. With the compression layer, less noise is required to hide the input, improving the utility. For a compression layer, we use a 1×1 convolution layer of $\phi: \mathbb{R}^{c_1 \times w \times h} \to \mathbb{R}^{c_2 \times w \times h}$ [27] for CNN, and a fully connected (FC) layer of $\phi: \mathbb{R}^{c_1} \to \mathbb{R}^{c_2}$ for MLP, where $c_1 > c_2$. Similarly, we used a 1×1 convolution layer of $\phi': \mathbb{R}^{c_2 \times w \times h} \to \mathbb{R}^{c_1 \times w \times h}$ [27] or a fully connected (FC) layer of $\phi': \mathbb{R}^{c_2} \to \mathbb{R}^{c_1}$ for decompression.

**Optimization 2: SNR loss** The second optimization directly adds a loss term during the model optimization so that the trained model requires less noise to be added to achieve the same dFIL. As the noise that needs to be added is $\sigma = \sqrt{\mathrm{Tr}(\mathbf{J}_{\mathcal{M}_{client}}^T \mathbf{J}_{\mathcal{M}_{client}})/(d * \mathrm{dFIL})}$ for a target dFIL, maximizing the signal-to-noise-ratio (SNR) of the split layer ($\mathbf{z}^T \mathbf{z}/\sigma^2$) is equivalent to minimizing $\mathrm{Tr}(\mathbf{J}_{\mathcal{M}_{client}}^T \mathbf{J}_{\mathcal{M}_{client}})/(\mathbf{z}^T \mathbf{z})$. We add the *SNR loss* term to the objective of the training, which trains the model parameters in a way that requires less noise to achieve the same dFIL. Our SNR loss is similar to Jacobian regularizer [37] proposed for robust learning. However, SNR loss differs in that it minimizes the Jacobian *normalized by the activation*.

## 5 Evaluation

Our evaluation aims to study whether dFIL and ReFIL can improve the privacy of split inference against input reconstruction attacks. Especially, we aim to answer the following questions: (1) Is dFIL a good measure for privacy?; (2) Can a compression layer improve the utility of ReFIL?; (3) Can an SNR loss improve the utility of ReFIL?

### 5.1 dFIL Correlates Well with the Input Reconstruction Quality

To understand the relationship between dFIL and the reconstructed input quality, we split popular networks, applied ReFIL with various target dFIL, and performed an input reconstruction attack. Evaluations in this section did not adopt a compression layer or an SNR loss.

**Unbiased attacker 1: image model** To evaluate the case of an unbiased attacker, we ran two sets of experiments. The first experiment was already presented in Figure 2. For the first experiment, we ran MNIST [22] dataset through a multi-layer perceptron (MLP). We split the model right after the

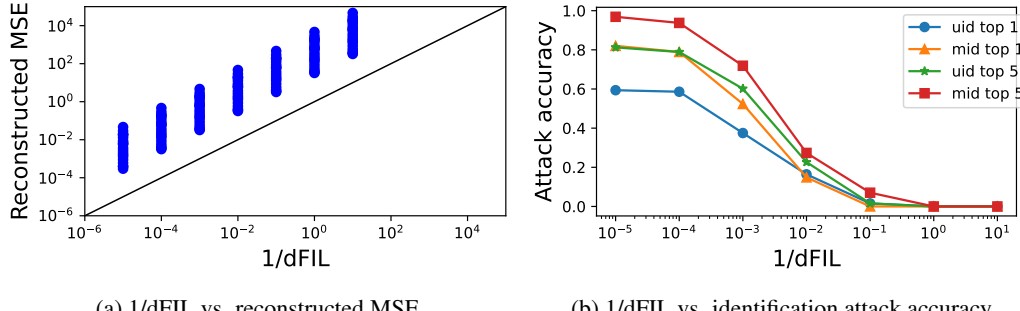

(a) 1/dFIL vs. reconstructed MSE

(b) 1/dFIL vs. identification attack accuracy

Figure 3: For an id reconstruction attack targeting recommendation models, 1/dFIL again (a) lower-bounds the reconstruction error of the embeddings, (b) controlling the attack accuracy.

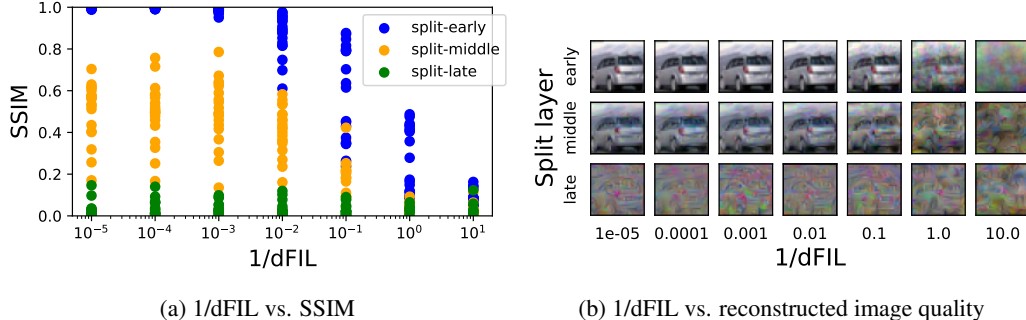

(a) 1/dFIL vs. SSIM

(b) 1/dFIL vs. reconstructed image quality

Figure 4: Even for biased attacks, dFIL correlates well with the reconstructed image quality, both (a) quantitatively and (b) qualitatively. SSIM is higher if the attack is more successful.

first FC layer, and tried reconstructing the input image from the split layer activation. For the attack method, we used a simple white-box reconstruction attack similar to [7, 9] that optimizes a randomly-initialized input to minimize the MSE of the split layer activation: $\hat{\mathbf{x}} = \arg\min_{\mathbf{x_0}} ||\mathbf{z}' - \mathcal{M}_{client}(\mathbf{x_0})||_2^2$.

As the optimization is convex, the attack should be unbiased. The attacker used Adam [38] with lr=0.1. As already discussed in Section 3, Figure 2 shows that *1/dFIL can lower-bound the average reconstruction MSE*, directly serving as a strong privacy metric.

**Unbiased attacker 2: recommendation model**  The second experiment used a MovieLens-20M [24] recommendation dataset and an MLP-based recommendation model [26]. The model first converts the user id (uid) and the movie id (mid) into an embedding representation using an embedding table, concatenates the embeddings, and passes it through an MLP to predict whether the user will like the movie [26]. We treated 5-star ratings as "like" and others as "non-like". The model used an embedding dimension of 32, and MLP layers of output size [64, 32, 16, 1]. The final ROC-AUC achieved without ReFIL was 0.8228.

To reconstruct the inputs (uid and mid) from the activation, we split the model after the first linear layer and used the same white-box attack as above to first reconstruct the embeddings. Then, uid/mid were reconstructed by finding the closest embedding value in the embedding table: $id = \arg\min_{i} ||\hat{emb} - Emb[i]||_2^2$, where $\hat{emb}$ is the reconstructed embedding and $Emb[i]$ is the $i$-th entry of the embedding table. Unbiased attacker 2 used the same hyperparameters as unbiased attacker 1.

Figure 3a plots the reconstructed MSE of the embeddings over different dFIL. Again, *1/dFIL correctly lower-bounds the reconstruction error*, being a good indicator for privacy. Figure 3b plots the final top-1 and top-5 attack success rate for different dFIL. As expected, the attack success rate is initially very high and decreases with decreasing dFIL. Perfect privacy is achieved near 1/dFIL $\geq$ 1.

**Biased attacker: image model + attacker with a prior**  When the attacker leverages prior knowledge of the input data, 1/dFIL cannot accurately lower bound the reconstruction MSE. For example,

| Setup | | 1/dFIL | No opt. | + comp | + comp + SNR loss |
|---|---|---|---|---|---|
| CIFAR-10 (base acc: 93.08%) | split-early | 1 | 10.26% | 10.65% (+3.8%) | **24.27% (+136.55%)** |
| | | 10 | 9.86% | 10.49% (+6.39%) | **13.57% (+37.63%)** |
| | | 100 | 9.97% | 9.92% (-0.5%) | **10.2% (+2.31%)** |
| | split-middle | 1 | 58.84% | 91.42% (+55.37%) | **91.95% (+56.27%)** |
| | | 10 | 17.75% | 85.00% (+378.87%) | **86.93% (+389.75%)** |
| | | 100 | 11.54% | 40.45% (+250.52%) | **47.73% (+313.6%)** |
| | split-late | 1 | 93.05% | **93.14% (+0.1%)** | 90.80% (-3.35%) |
| | | 10 | 91.33% | **92.57% (+1.36%)** | 90.24% (-1.19%) |
| | | 100 | 70.88% | **87.4% (+23.31%)** | 84.5% (+19.22%) |
| MovieLens-20M (base AUC: 0.8228) | | 1 | 0.8172 | 0.8291 (+1.46%) | **0.8294 (+1.49%)** |
| | | 10 | 0.7459 | 0.8211 (+10.08%) | **0.8223 (+10.24%)** |
| | | 100 | 0.6120 | 0.7640 (+24.84%) | **0.7675 (+25.41%)** |

Table 1: Adding a compression layer (+ comp) and an SNR loss (+ SNR loss) improves accuracy.

no matter how small dFIL is, if the attacker knows that the pixel value is always between [0, 1] and produces only values within that range, the reconstruction MSE will never go over 1. However, we show that dFIL still serves as a good indicator for privacy even in such cases.

For the biased attacker evaluation, we used CIFAR-10 [23] dataset and ResNet-18 [25]. We standardized the input image into $\mathcal{N}(0, 1)$ and used the same hyperparameters as in [39] to train the model for best accuracy, achieving 93.08% test accuracy. We explored three different splitting configuration, splitting early (*split-early*, right after the first convolution layer), at the middle (*split-middle*, after block 4), and late (*split-late*, after block 6). We used a similar white-box attacker as in the unbiased case, but with a total variation (TV) prior [7, 9] with $\lambda$=0.05. We also tried other biased attacks [8, 33], but omitted those results for brevity as the results were similar.

Figure 4a plots the structural similarity (SSIM) [40], a popular metric for image similarity, between the original and the reconstructed image over different dFIL. Smaller dFIL leads to lower SSIM, indicating the reconstructed image quality degrades with decreasing dFIL. Figure 4b shows that $1/\text{dFIL} \geq 10$ achieves perfect privacy across all setups, while others do or do not reveal private inputs depending on the split point. The privacy estimated by dFIL is sometimes conservative: for split-late, the attack was not able to reconstruct any input regardless of the dFIL value.

## 5.2 A Compression Layer and an SNR Loss Improve Utility

Table 1 shows the model accuracy after applying ReFIL with different dFIL, with and without the proposed optimizations. Without any optimizations (No opt.), ReFIL degrades the overall accuracy significantly when a low-enough dFIL (1/dFIL=1–100) is selected for strong privacy.

Adding a compression layer (+ comp) significantly improves the accuracy in almost all cases, sometimes up to 378.87%. Adding the SNR loss (+ SNR loss) on top gave an additional performance improvement for most of the cases, up to 389.75%. With the optimizations, some setups that were not practical due to too-low accuracy become practical. For example, CIFAR-10 + split-middle with 1/dFIL=1 experiences a boost in accuracy from an unreasonably-low 58.84% to a practical 91.95%. Similarly, 1/dFIL=10 for the same setup sees an accuracy improvement from 17.75% to 86.23%.

CIFAR-10 + split-early experiences too-low accuracy, even after the optimizations improve the accuracy by up to 136.55% (from 10.26% to 24.27% for 1/dFIL=1). The result shows that if we split too early, simply too much noise has to be added to hide the input and the system becomes impractical. We also see that CIFAR-10 + split-late sometimes shows a reasonable accuracy even before applying any optimizations (e.g., 93.05% when 1/dFIL=1), as ReFIL correctly captures the fact that the attack is already hard (Figure 4b) and only adds minimal noise.

## 6 Conclusion

Split inference and learning leak private information of the client through split layer activation. For the first time to the best of our knowledge, we propose a theoretically-meaningful metric, dFIL, and a privacy-enhancing method, ReFIL, to quantify and control the privacy leakage through the split layer, in the context of split inference and input reconstruction attacks. We show that dFIL and ReFIL is effective through a set of evaluations. We envision dFIL and ReFIL to be potentially extended to other setups where the model is split as well, such as split learning.

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

# A   Appendix

Optionally include extra information (complete proofs, additional experiments and plots) in the appendix. This section will often be part of the supplemental material.

