# OpenReview forum: "Measuring and Controlling Split Layer Privacy Leakage Using Fisher Information"
_NeurIPS.cc/2022/Workshop/Federated_Learning — FL-NeurIPS 2022 Poster_

### Official Review · Reviewer_7CEY · 2022-10-14
**a seems interesting paper**

This paper first proposes to use dFIL as a privacy metric for split layer activation. This paper then proposes ReFIL, a privacy-enhancing method that controls the information leakage of a split layer activation while maintaining reasonable utility through several optimizations. The effectiveness of dFIL and ReFIL are validated through a set of evaluations.

Strengths:
1. Information leakage and control is an interesting problem in split learning.
2. Overall structure is good.
3. Datasets are various, and writing is mostly clear.

Weaknesses:
1. the proposed privacy metric seems not general, only input reconstruction attacks were considered.
2. the system becomes perfectly private when 1/dFIL ≥ 1. I'm curious about its theory support, no any formal theory was given, all the experiments are empirical ones. However, this papers claims that dFIL is a "theoretically-meaningful metric", and "theoretically and empirically show that dFIL serves as a good privacy metric against input reconstruction attacks".
3. split learning has many types, I'm wondering whether the proposed dFIL is compatible with other types of split learning.

---

### Official Review · Reviewer_KKyd · 2022-10-16
**Comments**

In this work, the authors propose a metric, dFIL, and a privacy-enhancing method, ReFIL, to quantify and control privacy leakage through the split layer. They evaluate the efficacy of dFIL and ReFIL against input reconstruction attacks. The experimental results show that dFIL can be used as an indicator of privacy to provide an intuitive understanding of how much private information leaks. And ReFIL can also enhance privacy while maintaining utility.

Strengths

1.	This paper considers the problem of the privacy leakage of split layers, which is valuable and exists in many real-world scenarios.

2.	The authors leverage a method to quantitatively measure privacy leakage through input reconstruction attacks.

3.	The paper is easy to follow.

Weaknesses

1.	Do one need to run ReFIL for each input? If yes, please show the experiments of the computational complexity.

2.	It’s better to present a survey of FIM and show why you use dFIM instead of other ones that can be served as baselines in the experiments.

3.	Can you provide some theoretical results about the proposed defense? Or discuss the limitation of the method.

4.	Some content can be placed in the appendix. It is better to write the proof of the diagonal Fisher information leakage in the appendix. Similar experimental results in Section 5 can also be placed in the appendix.

Minor issues

1.	The figures and tables style can be optimized appropriately.

2.	The mix of dFIL and 1/dFIL is confusing.

---

### Official Review · Reviewer_vGQS · 2022-10-17

This paper proposes to use Fisher information as a privacy metric to measure and control information leakage. This paper also proposes a privacy-enhancing technique to prevent the reconstruction attack at the split layer. This paper conducted several experiments to support their claims.

However, I still have several questions/concerns.

1. in comparison with previous work such as [21], what's the novelty of this paper?

2. Since privacy is preserved by adding Gaussian noise, what's the difference between ReFIL and differential privacy? any theoretical and experimental comparisons?

3. Missing comparison partners and discussion of relevant work. such as: [a] Defending against Reconstruction Attack in Vertical Federated Learning; [b] Label Leakage and Protection from Forward Embedding in Vertical Federated Learning. Both paper talked about information leakage and protection from the split layer.

---

### Decision · Program_Chairs · 2022-10-20

Accept (Poster)